# Apoptosis in Cardiac Conditions Including Cirrhotic Cardiomyopathy

**DOI:** 10.3390/ijms26136423

**Published:** 2025-07-03

**Authors:** Fengxue Yu, Dae Gon Ryu, Ki Tae Yoon, Hongqun Liu, Samuel S. Lee

**Affiliations:** 1Liver Unit, University of Calgary Cumming School of Medicine, Calgary, AB T2N 2T8, Canada; 27400757@hebmu.edu.cn (F.Y.); gon22gon@naver.com (D.G.R.); ktyoon@pusan.ac.kr (K.T.Y.); hliu@ucalgary.ca (H.L.); 2Second Hospital of Hebei Medical University, Shijiazhuang 050004, China; 3Division of Gastroenterology, Yangsan Hospital, Pusan National University Faculty of Medicine, Busan 46241, Republic of Korea

**Keywords:** cirrhosis, apoptosis, heart diseases, cirrhotic cardiomyopathy, treatment

## Abstract

Apoptosis is a highly regulated process of programmed cell death and plays a crucial pathogenic role in a variety of conditions including cardiovascular diseases. There are two pathways leading to apoptosis, the intrinsic and extrinsic pathways. In the intrinsic pathway, also known as the mitochondria-mediated pathway, the cell kills itself because it senses cell stress. Mitochondria account for 30% of cardiomyocyte volume, and therefore, the heart is vulnerable to apoptosis. The extrinsic pathway, also known as the death receptor-mediated pathway, is initiated by death receptors, members of the tumor necrosis factor receptor gene superfamily. Excessive apoptosis is involved in cardiac dysfunction in different cardiac conditions, including heart failure, ischemic heart disease, and cirrhotic cardiomyopathy. The last entity is a serious cardiac complication of patients with cirrhosis. To date, there is no effective treatment for cirrhotic cardiomyopathy. The conventional treatments for non-cirrhotic heart failure such as vasodilators are not applicable due to the generalized peripheral vasodilatation in cirrhotic patients. Exploring new approaches for the treatment of cirrhotic cardiomyopathy is therefore of utmost importance. Since apoptosis plays an essential role in the pathogenesis and progression of cardiovascular conditions, anti-apoptotic treatment could potentially prevent/attenuate the development and progression of cardiac diseases. Anti-apoptotic treatment may also apply to cirrhotic cardiomyopathy. The present review summarizes apoptotic mechanisms in different cardiac diseases, including cirrhotic cardiomyopathy, and potential therapies to regulate apoptosis in these conditions.

## 1. Introduction

Apoptosis, a form of programmed cell death, is a highly regulated process that allows a cell to self-degrade in order for the body to eliminate any unwanted or dysfunctional cells. This process involves *DNA* fragmentation, and *mRNA* decay. Other changes involve the organelles, cell membrane, and nucleus. The morphological changes include blebbing, chromatin condensation, nuclear fragmentation, cell contraction, and the disintegration of part of the cell to form smaller apoptotic bodies [1].

Cell death can occur by programmed death such as apoptosis, and non-programmed death such as necrosis. There are important differences between these two processes. Apoptosis is an active, programmed process with a series of molecular steps in a cell that leads to cell death. In comparison, necrosis is a passive, accidental cell death due to uncontrolled environmental perturbations [2].

The functions of apoptosis can be pathological, which causes damage and results in diseases, or physiological, a process essential for normal health. The physiological roles of apoptosis are mainly eliminating unwanted cells. An example is the developing process of digits in embryos, where the cells joining the fused digits die and slough off, leaving individual digits [3]. An example after birth is the wound healing process, where apoptosis removes the inflammatory cells and granulation tissue [4]. In adults, apoptosis helps maintain homeostasis in organisms: the rate of cell growth and death is balanced to maintain organ sizes, functions, and maintain body weight. Abnormalities of apoptosis, either insufficient or excessive, can cause multiorgan diseases such as peptic ulcer [5], and liver, heart, lung and kidney diseases, as well as cancers [6].

Cardiac function in patients with cirrhosis manifests normally at rest due to a decreased peripheral vascular resistance. However, when challenged, the systolic and diastolic contractile responses are blunted. Moreover, morphological changes, including enlargement or hypertrophy of cardiac chambers, and electrophysiological repolarization changes, including a prolonged QT interval, are observed. The constellation of these cardiac abnormalities is called cirrhotic cardiomyopathy [7]. The present review focuses on the role of apoptosis in cardiac diseases including cirrhotic cardiomyopathy.

## 2. Disordered Cardiovascular (CV) System in Cirrhosis

Cirrhosis is the final stage of chronic liver disease, which is characterized by an irreversible substitution of fibrotic tissue and regenerative nodules for normal liver parenchyma [8].

Because of the fibrosis and pseudolobule formation, the vasculature of the liver is distorted, which impedes the circulation and increases the resistance to the blood flow through the organ. All these changes result in portal venous hypertension.

Portal hypertension causes intestinal congestion, which results in bacterial translocation, bacterial overgrowth, and an increase in intestinal permeability. These changes then lead to the release of endotoxin into the circulation. The increased circulating endotoxin stimulates pro-inflammatory cytokines such as TNFα and interleukin 1β [9]. These inflammatory cytokines further increase the production of vasodilators such as nitric oxide (NO) and carbon monoxide (CO) [10]. The vasodilators decrease peripheral vascular resistance and blood pressure, and increase cardiac output. This constellation of phenomena is termed hyperdynamic circulation [11].

Another disordered cardiovascular phenomenon is heart dysfunction, which features as increased cardiac output at baseline, as well as blunted ventricular inotropic and chronotropic responses to stimuli such as exercise, drugs, hemorrhage and surgery. This condition is called cirrhotic cardiomyopathy (CCM) [12]. Cardiomyopathies in non-cirrhotic patients include dilated cardiomyopathy, hypertrophic cardiomyopathy, and restrictive cardiomyopathy. Cirrhotic cardiomyopathy more resembles a hypertrophic cardiomyopathy as there is left ventricular hypertrophy in cirrhotic animal models [13]. The clinical features and diagnostic criteria of CCM have recently been reviewed in detail [12]. The updated diagnostic criteria were formulated by an expert consensus working group, the Cirrhotic Cardiomyopathy Consortium (Table 1). This condition affects approximately 20–30% of patients with advanced cirrhosis, and is clinically significant because it contributes to the genesis of acute kidney injury/hepatorenal syndrome, as well as worse outcomes following interventions such as insertion of transjugular intrahepatic portosystemic shunts (TIPSs) and liver transplantation [12].

It is now clear that the pathogenesis of cirrhotic cardiomyopathy includes cardiac inflammation, manifested as increased inflammatory cells [14] and inflammatory cytokines [15]; increased cardiac oxidative stress, manifested as increased reactive oxidative species (ROS) and decreased antioxidants [16]; and increased apoptosis [17].

## 3. Apoptosis as a Factor in Cardiac Dysfunction

Heart diseases are global health issues that remain the top cause of mortality. Cardiomyocytes are the main functional cells of cardiac contraction and are essential for maintaining the normal pumping function of the heart. One of the key cellular processes related to cardiomyopathy is cardiomyocyte death. Because cardiomyocytes are terminally differentiated with limited regenerative capacity, cardiomyocyte death is irreversible [18,19]. Continuing damage to cardiomyocytes leads to the progression of cardiac dysfunction (Table 2). The majority of cardiomyocyte cell loss is regulated cell death, including apoptosis, necroptosis, mitochondrial permeability transition pore (mPTP) necrosis, ferroptosis, and pyroptosis [19].

Cardiomyocyte apoptosis occurs in many cardiovascular diseases [18]. Wencker and colleagues have shown that only a 0.023% rate of cardiomyocyte apoptosis is sufficient to induce a lethal dilated cardiomyopathy in mice [24]. Regulated forms of cell death play crucial roles in a variety of cardiac conditions such as ischemia/reperfusion injury in myocardial infarction, heart failure, and doxorubicin-induced cardiotoxicity [25]. More than a dozen cell death programs have been described such as necroptosis, mitochondrial permeability transition pore (mPTP)-dependent necrosis, ferroptosis, and pyroptosis [26]. All forms of excessive cardiomyocyte death can contribute to the pathogenesis of cardiac diseases [27].

## 4. Other Types of Cell Death

Necroptosis: Membrane-bound death receptors (DRs) are the initiators of necroptosis. DRs, via the homologous receptor interacting protein kinase 1 (RIPK1), phosphorylates and activates receptor interacting protein kinase 3 (RIPK3). The activated RIPK further phosphorylates and activates mixed lineage kinase-like domain (MLKL). RIPK1 and RIPK3 together with MLKL form complex IIc, also known as the necrosome, which induces necroptotic cell death. The necrosome was upregulated in failing human hearts compared with healthy controls [28]. Ablation of the RIPK3 gene improves cardiac function and survival in a murine model of myocardial infarction 33854696, which suggests that RIPK3-dependent necroptosis plays an important pathogenic role in heart failure [29].

Ferroptosis: Ferroptosis, also called oxytosis, is a mode of non-programmed cell death dependent on iron and acting via lipid peroxidation. Ferroptosis of cardiomyocytes appears to be a key pathological mechanism of heart failure [30]. Factors such as pressure overload, inflammatory infiltration and oxidative stress can induce ferroptosis 40360081. Therapeutic strategies targeting ferroptosis such as an iron chelator ferrenostatin-1 improve cardiac function in heart failure [31].

Pyroptosis: Pyroptosis is a type of inflammation-driven programmed cell death [31], characterized by simultaneous apoptosis and necrosis. The destruction of the cell membrane causes the release of pro-inflammatory factors such as IL-1β and IL-18 to the extracellular space, which triggers inflammatory reactions and, consequently, cardiac dysfunction [32]. Pyroptosis plays a crucial role in the pathogenesis of cardiac diseases and ultimately, heart failure [33]. Different therapeutic strategies that inhibit pyroptosis have been demonstrated to improve cardiac structure and function [33].

The literature shows that all modes and pathways that lead to the death of cardiomyocytes impair cardiac function. Moreover, there are interactions amongst the various types of cardiomyocyte death [30]. However, to date, only the impact of apoptosis on cirrhotic cardiomyopathy has been studied [17], and therefore, the impact of each/or combination of different types of cardiomyocyte death needs further research.

## 5. Heart Failure

Heart failure is defined as the failure of the heart to pump sufficient blood to support the tissues at rest or during activities. Heart failure has a high incidence worldwide with significant morbidity and mortality [34]. Cardiomyocyte apoptosis is an important mechanism for the transition from myocardial hypertrophy or dilatation to heart failure [20]. The clinical features include pressure overload, volume overload [35], and insufficient blood supply [35]. Heart failure is the ultimate outcome of different cardiac diseases, and its pathogenesis includes cardiac oxidative stress, inflammation, and cardiomyocyte loss.

Cardiomyocyte apoptosis is one of the important causes of heart failure [18]. Using rapid cardiac pacing to create a heart failure model in rabbits, Qin et al. [21] showed that the number of apoptotic myocytes was significantly increased in cardiac muscle from rabbits with heart failure. They further noticed that in the failing hearts, the anti-apoptosis factor Bcl-2 was significantly decreased. On the other hand, the pro-apoptosis factor Bax was significantly increased, and consequently the ratio of Bcl-2 to Bax was tremendously reduced. Interestingly, the number of apoptotic myocytes was negatively correlated with the cardiac contractile function (with maximal rate of rise of left ventricular pressure, *dp/dt*, r = 0.605, *p* < 0.001, and fractional shortening, r = 0.741, *p* < 0.001). Furthermore, selegiline, a monoamine oxidase inhibitor, significantly decreased the number of apoptotic myocytes, increased Bcl-2, decreased Bax, and increased the ratio of Bcl-2 to Bax in failing hearts. In parallel with the effects of selegiline to reduce apoptosis, a significant improvement in cardiac contractility was observed, manifested as a decreased left ventricular end-diastolic pressure, increased left ventricular *dp/dt*, and fractional shortening. These data strongly suggest that cardiomyocyte apoptosis directly impacts cardiac contractility.

Another study by Sun and coworkers [20] used isoproterenol to induce heart failure in mice and observed the existence of apoptotic cardiomyocytes under electron microscopy. They also noted that isoproterenol significantly decreased the expression of Bcl-2 and increased the expression of Bax and cleaved caspase-3, the ‘executor of apoptosis’ [36] in cardiac tissue. Ivabradine, a pacemaker current inhibitor, decreased the expression of Bax and cleaved caspase-3 protein, increased the expression of Bcl-2, and inhibited cardiomyocyte apoptosis. Moreover, besides the decreased heart rate in the failing heart, ivabradine also significantly improved cardiac function in isoproterenol-induced heart failure, with increased left ventricular ejection fraction (LVEF) and fractional shortening.

## 6. Myocardial Infarction

Myocardial infarction is still a leading cause of morbidity and mortality worldwide. This devastating condition results from the occlusion of a coronary artery, which reduces the blood flow and oxygen supply to a segment of the myocardium. This situation further causes irreversible cellular damage and cardiomyocyte death with the potential for adverse cardiac remodeling and impaired cardiac function [37]. Myocardial infarction is featured by a significant loss of cardiomyocytes [22], which causes cardiac dysfunction. Therefore, salvaging cardiomyocytes after infarction is critical for improving ventricular function and prognosis.

There are several ways to alleviate the apoptosis of cardiomyocytes and improve cardiac function, including interfering with pro-apoptotic protein expression such as by miRNA [38,39], and chemical interventions such as galectin-3 inhibitor and thioredoxin. Ibarrola and coworkers [23] used coronary artery ligation to create ischemia–reperfusion (IR) model in rats and showed that galectin-3 was significantly increased in the ischemic area. It was previously demonstrated that galectin-3 inhibition plays a significant role in protecting against cardiomyocyte apoptosis [40,41]. Ibarrola et al. also reported that modified citrus pectin treatment decreased galectin-3 expression and the size of the ischemic area. Furthermore, modified citrus pectin decreased levels of inflammatory markers such as IL-β and C-reactive protein (CRP), and also B-type natriuretic peptide (BNP), a marker of heart dysfunction. Galectin-3 blockade was also associated with less myocardial inflammation and fibrosis. These data show the therapeutic potential of galectin-3 inhibition on apoptosis.

Another study [42] demonstrated that galactin-3 was significantly increased in the myocardium and cardiomyocyte in rats subjected to ischemia–reperfusion injury. Galectin-3 knockdown significantly mitigated the apoptosis of cardiomyocytes and the size of the ventricular ischemic area. An investigation of the mechanistic pathway revealed that galectin-3 interacted with bcl-2. Galectin-3 binding to bcl-2 blocked the anti-apoptotic effect of bcl-2 and enhanced the myocardial apoptosis. The authors therefore concluded that galectin-3 knockdown prevented myocardial ischemia–reperfusion injury through the release of bcl-2.

Another anti-apoptotic agent is thioredoxin. Medali and coworkers [22] used left anterior descending coronary artery permanent ligation to create myocardial infarction in mice to test the effects of thioredoxin-1 (Trx-1), Trx-80, and mimetic peptide of Trx-1 (Ac-Cys-Pro-Cys-amide, CB3) on cardiomyocyte apoptosis and cardiac function. They showed that CB3 was more potent than Trx-1. CB3 significantly increased the expression of the anti-apoptotic protein, Bcl-2 (1.43 ± 0.03 vs. 0.63 ± 0.20, *p* = 0.01), and decreased the proapoptotic protein, Bax (0.14 ± 0.02 vs. 0.32 ± 0.05, *p* = 0.04), in myocardial infarction compared with controls. It also significantly decreased the Bax/Bcl-2 ratio (0.11 ± 0.01 vs. 0.72 ± 0.13, *p* = 0.003). TUNEL labeling showed that CB3 significantly alleviated apoptosis (11.38 ± 0.43 for CB3 vs. 27.08 ± 0.95 for control, *p* = 0.02). All these effects resulted in the reduction in cardiac infarct size in the group of myocardial infarction + CB3 compared with myocardial infarction controls. Besides the effects of CB3 on the apoptosis of cardiomyocytes, it also significantly alleviated cardiac inflammation and oxidative stress.

The study by Medali and colleagues on cardiac function showed that CB3 significantly increased LVEF (58.45% ± 1.92% vs. 36.44% ± 3.29%, *p* = 0.0001) and fractional shortening (28.78% ± 1.06% vs. 15.69% ± 1.52%, *p* = 0.0001), and significantly reduced the end-systolic left ventricular diameter (0.29 ± 0.01 vs. 0.42 ± 0.02, *p* < 0.0001). All these data demonstrated the relationship between the apoptosis of cardiomyocytes and cardiac function.

## 7. Role of Apoptosis in Cirrhotic Cardiomyopathy

Cirrhotic cardiomyopathy is defined as systolic and diastolic dysfunction, electrophysiological changes, and macroscopic structural changes [43]. Many hypotheses have been proposed to elucidate the pathogenesis of cirrhotic cardiomyopathy. However, the exact mechanism of cirrhotic cardiomyopathy remains unclear.

Theoretically, apoptosis, inflammation, and oxidative stress are three components that could cause cardiac injury and dysfunction. These three components rely on each other and have additive/synergistic effects. Apoptosis, one of the important components in this pathogenic triad, plays an important role in cirrhotic cardiomyopathy [17]. Many factors that underlie cirrhotic cardiomyopathy are associated with apoptosis.

## 8. Oxidative Stress and Apoptosis

There is a dynamic relationship between reactive oxygen species (ROS) production and antioxidant capacity of any given cell system. In physiological status, the pro-oxidant-to-antioxidant ratio stays balanced. When pro-oxidants become dominant in the system, tissue damage occurs [44]. Oxidative stress is defined as “a serious imbalance between the generation of ROS and antioxidant defenses in favor of ROS, causing excessive oxidative damage” [45]. Oxidative stress is a consequence of an increased generation of free radicals and/or reduced physiological activity of antioxidant defenses against free radicals, which occurs when oxygen radical production surpasses the cell’s capacity of detoxification [46]. ROS are generated in cells by both exogenous and endogenous stimuli. Endogenous ROS are stimulated by both inflammatory cytokines and leaks during mitochondrial electron transport chain activity [47].

There is a cause-and-effect correlation between oxidative stress and apoptosis. The mechanisms are multifactorial. Oxidative stress damages mitochondrial function which dramatically decreases cellular energy supply, thus leading to apoptosis. Besides the depletion of energy supplies, ROS react with multiple biological macromolecules, such as lipids, proteins, nucleic acids, and carbohydrates. These reactions damage the organelles of the cells, leading to activation of cell death processes [48].

It is well known that mitochondria are the main sites that generate ROS, and the heart is enriched with mitochondria (30% of cardiac cell volume). Therefore, the heart is vulnerable to ROS-mediated damage [49]. Oxidative stress plays important roles in non-cirrhotic heart diseases such as ischemic cardiac diseases [50] and heart failure [51]. In patients with cirrhosis, pro-oxidants overwhelm antioxidants. Thus, oxidative stress is a key pathogenic factor in chronic liver injury of various etiologies [52]. We previously demonstrated in rats that oxidative stress is a major culprit responsible for the development of hyperdynamic circulation in portal hypertension [53]. Furthermore, oxidative stress also plays a significant role in cirrhotic cardiomyopathy [16]. In cirrhosis, portal venous hypertension causes mesenteric congestion, which stimulates the production of ROS. Overproduction of ROS results in intestinal mucosal injury, increased intestinal permeability, and gut bacterial overgrowth and translocation to the systemic circulation. All these factors alter the intestinal environment and increase endotoxinemia and inflammation, which in turn enhances oxidative stress. This cycle further worsens cardiac function. Blocking oxidative stress can minimize the systemic inflammatory response and alleviate the severity of cirrhotic cardiomyopathy [52]. From the evidence available in the literature, we speculated that cardiomyocyte apoptosis may be due to excessive oxidative stress in cirrhotic cardiomyopathy.

## 9. Inflammation

Cirrhotic cardiomyopathy manifests as an inflammatory phenotype, i.e., inflammatory cell infiltration [14] and increased pro-inflammatory cytokines such as TNFα and interleukin 1β [9]. We demonstrated that the increased monocyte recruitment in the myocardium in bile duct ligated (BDL) rat heart decreases myocardial contractility [14], an effect partially mediated by an increased production of pro-inflammatory cytokines such as TNFα.

## 10. TNFα

It is well documented that TNFα plays a significant role in cardiac dysfunction. The mechanisms are multifaced, including providing an inflammatory microenvironment that impacts cardiac function [54] via the NF-κB-iNOS and p38MAPK signaling pathways [15], increasing oxidative stress [15,55], and by aggravating the apoptosis of cardiomyocytes.

The TNF receptor superfamily includes the TNF receptor superfamily member 1A (TNFR1) and TNF-related apoptosis-inducing ligand receptors (TRAIL-Rs), including TRAILR1 and TRAILR2 [56]. Increased expression of TRAILs increases apoptosis in the heart and the TRAIL blockade could offer potential cardioprotection in the setting of myocardial infarction [57]. The binding of TNFα to TNFR1 recruits TNF receptor-associated death domain (TRADD), and thus, complex I is formed. Complex I transits to complex II, which includes complex I endocytosis, TNFR1 dissociation, RIP1 (receptor-interacting protein 1), deubiquitination, and FADD and procaspase-8 recruitment [58]. Procaspase-8 further causes apoptosis via death effector caspase 3 and caspase 7 [59]. In summary, TNFα is a pro-apoptotic factor that induces apoptosis via a complex pathway.

TNFα is a contributor to apoptosis in cardiovascular diseases. TNFα induces apoptosis in both myocytes and endothelial cells [60]. Bryant et al. [61] used transgenic mice to overexpress TNFα in cardiomyocytes in vivo, and showed that these mice developed a severe impairment of cardiac function manifested as biventricular dilatation and depression of ejection fraction, leading to premature death. The pathological examination showed that overexpression of TNFα caused cardiomyocyte apoptosis. Sallberg et al. [62] documented that high levels of plasma TNFα in patients with left ventricular heart failure was associated with myocardial apoptosis and the worsening of heart failure.

Moe and co-workers [63] used chronic rapid pacing in dogs to induce a heart failure model which mimics dilated cardiomyopathy in humans [64]. They demonstrated that TNFα was significantly increased in both sera and heart tissue of the heart failure group, and these cardiomyocytes showed reduced activity of the mitochondrial respiratory chain enzyme complex. Moreover, the apoptotic positive cells in cardiac tissue from the failing hearts were significantly increased. The application of etanercept, an inhibitor of TNFα, significantly increased the activity of the mitochondrial respiratory chain enzyme complex and decreased apoptotic positive cells and DNA fragmentation. In the evaluation of cardiac function, Moe et al. found that left ventricular ejection fraction in the failing hearts significantly decreased, left ventricular end-diastolic volume was significantly increased, and the inhibition of TNFα partially reversed these changes. These data demonstrate that TNFα is correlated to the impairment of mitochondrial function and myocyte apoptosis. Furthermore, TNFα-induced apoptosis is associated with LV dysfunction in pacing-induced heart failure.

The correlation between TNFα and cirrhotic cardiomyopathy has been well documented. Using BDL to create cirrhotic models in both rats and mice, we demonstrated that cardiac TNFα content was significantly increased in BDL animals [9,15,16]. We used either TNFα knockout (TNFα-/-) to diminish TNFα content in both blood and cardiac tissue or infusion of anti-TNFα antibody to neutralize TNFα in vivo and showed that the reduction in TNFα significantly improved the cardiac contractile and relaxation functions in cirrhotic hearts [15].

## 11. Galectin-3

Galectin-3 plays an important role in cardiomyocyte apoptosis, and its inhibition protects cardiomyocytes against the apoptosis [40,41]. Using BDL to create cirrhotic cardiomyopathy in rats, we recently investigated the possible pathogenic role of galectin [65]. We divided the rats into four groups: BDL, BDL + N-acetyllactosamine (N-Lac, an inhibitor of galectin-3), sham-operated controls, and sham + N-Lac. We found that cardiac galectin-3 was significantly increased in the BDL-cirrhotic rats compared with sham controls. N-Lac had no effect on galectin-3 content in sham-control hearts. Interestingly, cardiac galectin-3 content was significantly lower in the BDL + N-Lac group compared with that in the BDL group. Another cardiac function index, brain natriuretic peptide (BNP) level, was significantly increased in BDL hearts compared with sham controls. N-Lac significantly decreased BNP content in cirrhotic hearts but not in sham-operated controls. Another interesting finding was that changes in TNFα levels paralleled the changes in galectin-3 and BNP. Furthermore, the cardiomyocyte systolic contractile and diastolic relaxation velocities were significantly decreased from BDL hearts compared with those of sham controls, and N-Lac significantly improved these contractile indices in BDL hearts but had no impact on sham control cardiomyocytes. Since galectin-3 is a pro-apoptotic factor, the impact of galectin-3 on cardiac function might be at least partially mediated through an effect on apoptosis. The results of this animal study might also eventually be applicable to patients [66].

In cirrhotic cardiomyopathy, both intrinsic and extrinsic pathways of apoptosis may participate in the pathogenesis (Figure 1). We created a mouse model of cirrhotic cardiomyopathy by BDL to evaluate the intrinsic and extrinsic pathways [17]. We mainly tested Bcl-2 (B-cell lymphoma 2, an anti-apoptosis protein), Bax (a pro-apoptosis protein), and the Bcl-2/Bax ratio in the intrinsic pathway. Western blot analyses showed that Bcl-2 was increased in BDL hearts, and the Bcl-2/Bax ratio was elevated, which was unexpected because the intrinsic pathway had a protective effect on apoptosis in cardiomyocytes. When we evaluated the extrinsic pathway, we found that Fas expression was significantly increased in the BDL heart (Figure 2). Since Fas receptor is a death receptor on the surface of cells, Fas is therefore a pro-apoptotic protein. Fas ligand (FasL), also a pro-apoptotic factor, was also increased in the BDL heart. Furthermore, FLIP (FLICE inhibitory protein), an anti-apoptosis protein, was significantly decreased in the BDL heart. The increase in Fas and decrease in FLIP expression in the heart of the BDL-mice strongly support the notion that the alteration of the extrinsic pathway favored apoptosis.

A key question is whether there is indeed an overall net pro-apoptotic imbalance in the BDL mouse heart. The intrinsic and extrinsic pathways in apoptosis finally converge in the activation of caspase-3, and the activated caspase-3 catalyzes PARP-1 (Poly ADP-ribose Polymerase-1) to 89 and 28 kDa fragments, which represents direct evidence of ongoing apoptosis [67,68]. We therefore evaluated PARP-1. Our immunohistochemistry showed a significant increase in PARP staining (mean number of positive nuclei per high power field, x400) on a slice from BDL hearts compared with sham controls (18.2 ± 11.4 vs. 6.7 ± 5.3; *p* < 0.05). We therefore confirmed that apoptosis indeed occurs in the BDL heart, and the extrinsic pathway dominates this process. The increases in Bcl-2 and Bcl-2/Bax ratio in the BDL heart reflects an anti-apoptotic compensatory reaction, but the net pro-apoptotic imbalance is mediated by the extrinsic pathway.

After the validation of the existence of apoptosis in the BDL mouse heart, we injected anti-FasL monoclonal antibody to BDL-mice to neutralize the Fas ligand and showed that this antibody not only significantly decreased FasL in the BDL heart, but also improved the attenuated cardiac contractile and relaxation velocities (Figure 3) [17]. With these data, we confirmed that cardiac apoptosis plays a significant pathogenic role in cirrhotic cardiomyopathy.

## 12. Potential Therapies to Mitigate Apoptosis in Cirrhotic Cardiomyopathy

Liver transplantation is a definitive curative treatment for cirrhotic cardiomyopathy. However, it remains unavailable or difficult to access in many regions worldwide because of the high costs, the shortage of donor organs, and the need for sophisticated infrastructure and expertise in surgical, anesthetic, and intensive care systems. Even in economically developed regions such as North America, where financial resources and the requisite expertise/infrastructure of support specialties is available, the critical shortage of donor organs hampers widespread use of this therapeutic modality. The conventional treatment for non-cirrhotic heart failure is not applicable due to the significant baseline vasodilatation in patients with cirrhosis. Therefore, exploring new strategies to treat patients with cirrhotic cardiomyopathy is crucial.

Apoptosis plays a key role in the pathogenesis and progression of cardiovascular disease [69]. Since the rate and amount of death of cardiomyocytes is likely the most important determinant of patient morbidity and mortality, therapeutic strategies targeting apoptosis are crucial [69].

Among the multiple factors that are involved in cirrhotic cardiomyopathy, three components—inflammation, oxidative stress, and apoptosis—play important roles. These three factors compose a ‘vicious cycle’, and therapies aim to block one of them to impede this cycle and improve cardiac function. Our previous reviews demonstrated the therapeutic strategies of modifying inflammation or oxidative stress on the treatment of cirrhotic cardiomyopathy [52,70]. There is no direct study of potential therapies to regulate apoptosis in cirrhotic cardiomyopathy. However, any effective treatment targeting apoptosis in non-cirrhotic heart dysfunction may also be applicable to cirrhotic cardiomyopathy.

There are many studies on the treatment of cardiac diseases by targeting apoptosis. Under the combined search terms of “heart, apoptosis, treatment”, there are >15,700 publications in PubMed to date, including polypeptides such as humanin [69] and thioredoxin [22]; miRNAs such as miRNA-21 [71], miRNA-144 [72], and miRNA 132; and chemicals such as empagliflozin [73], quercetin [74], and β-blockers [75]. All these approaches attenuate apoptosis in cardiac diseases and improve cardiac function. However, the effects of these approaches on cirrhotic cardiomyopathy need further study.

There is currently no definitive medical treatment of cirrhotic cardiomyopathy. Non-selective beta-blockers (NSBBs), which decrease portal vein pressure, mitigate intestinal congestion, and reduce absorption of endotoxin, may alleviate cardiac inflammation, oxidative stress, and apoptosis, but results to date using NSBBs in CCM have not shown significant efficacy [76]. Albumin, taurine, spermidine, and statin drugs also have anti-inflammatory and antioxidant effects and may improve cardiac function in cirrhotic cardiomyopathy, but efficacy has not yet been demonstrated in clinical trials of patients with CCM [76].

In conclusion, apoptosis plays an important role in cardiac diseases, including cirrhotic cardiomyopathy. To date, there is no validated treatment for cirrhotic cardiomyopathy. Some approaches targeting apoptosis may open a window for the treatment of patients with cirrhotic cardiomyopathy.

## Figures and Tables

**Figure 1 ijms-26-06423-f001:**
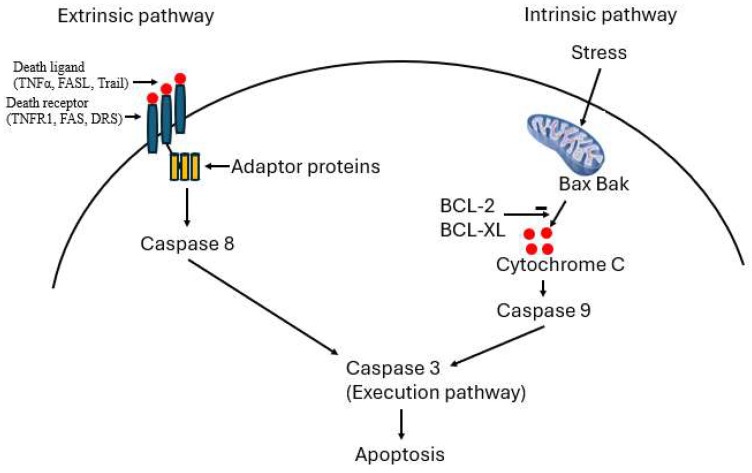
Extrinsic and intrinsic pathways in apoptosis.

**Figure 2 ijms-26-06423-f002:**
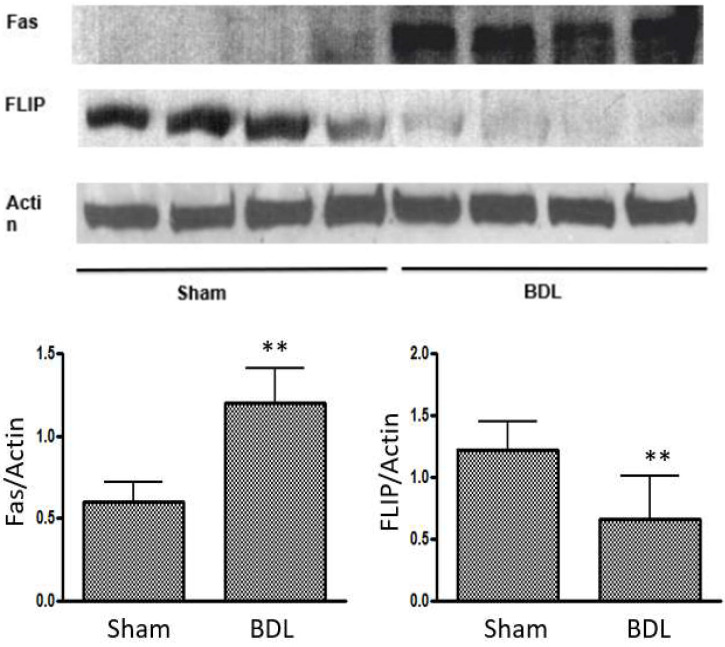
Representative Western blot analysis of Fas and FLIP protein expression in BDL-mice and sham control hearts. Lanes 1–4, sham-operated controls; lanes 5–8, BDL-mice. Computerized optical densitometry scanning showed that Fas was significantly increased and FLIP was significantly decreased in BDL-mice compared with their sham controls (** *p* < 0.05) (reproduced from [14]).

**Figure 3 ijms-26-06423-f003:**
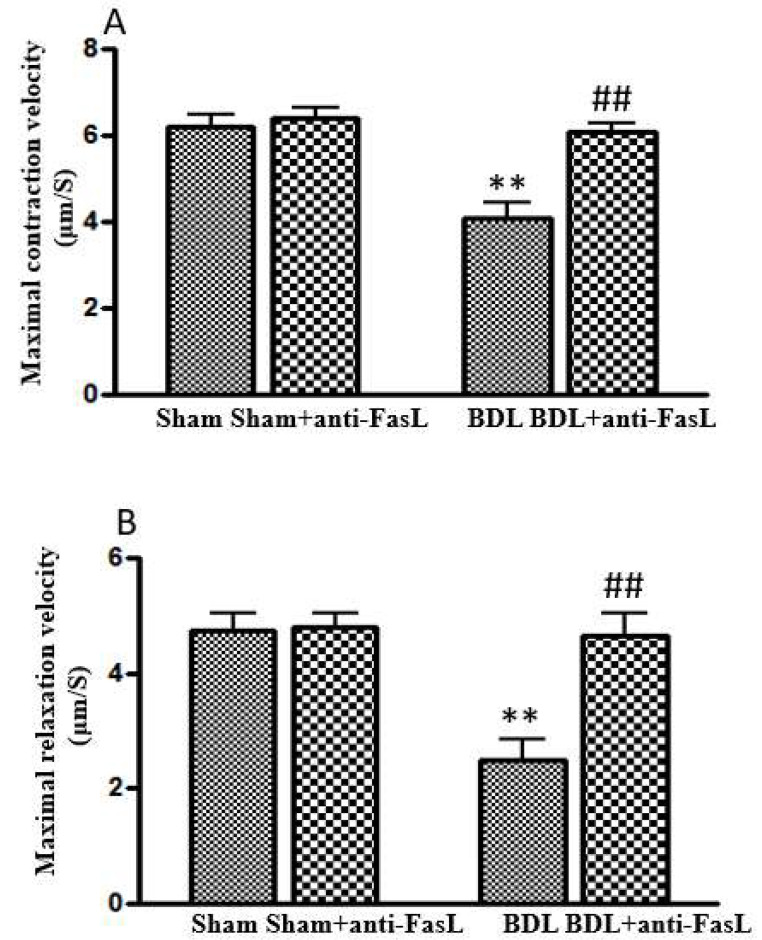
Isoproterenol-stimulated maximal systolic contraction and diastolic relaxation velocity in isolated cardiomyocytes. (**A**) Maximal systolic contraction velocity. (**B**) Maximal diastolic relaxation velocity. Both measures were significantly slower in BDL-mice compared with sham controls (** *p* < 0.01). Treatment of BDL-mice with anti-FasL monoclonal antibody significantly reversed the reduced systolic contraction and diastolic relaxation velocity in BDL-mice (## *p* < 0.01 compared with BDL) (reproduced from [14]).

**Table 1 ijms-26-06423-t001:** Diagnostic criteria for cirrhotic cardiomyopathy endorsed by the Cirrhotic Cardiomyopathy Consortium.

Criteria	Systolic Dysfunction	Diastolic Dysfunction
CCC criteria (2019)	LVEF ≤ 50% Or GLS < 18%	≥3 of the following: E/e’ ratio ≥ 15 e’ septal < 7 cm/s TR velocity > 2.8 m/s LAVI > 34 mL/m^2^

LVEF: left ventricular ejection fraction; GLS: global longitudinal strain; E/e’ ratio: ratio of early diastolic mitral inflow velocity to early diastolic mitral annulus velocity; TR: tricuspid regurgitation; LAVI: left atrial volume index.

**Table 2 ijms-26-06423-t002:** Role of apoptosis in non-cirrhotic cardiac conditions.

First Author (Ref.)	Subject	Disease/Model	Effects
Sun [20]	Mouse	Isoprenaline-induced heart failure	Ivabradine decreases apoptosis and improves cardiac function
Qin [21]	Rabbit	Pacing-induced heart failure	Selegiline decreases apoptosis and improves cardiac function
Medali [22]	Mouse	Coronary artery ligation-induced MI	Thioredoxin decreases apoptosis and improves cardiac function
Ibarrola [23]	Rat	coronary artery ligation-induced ischemia–reperfusion	MCP decreases galectin-3 and apoptosis, and improves cardiac function

MCP: Modified citrus pectin.

## Data Availability

Not applicable—no original data was shown.

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
