# Peer review of "Apoptosis in Cardiac Conditions Including Cirrhotic Cardiomyopathy"

_ijms, 2025, doi:10.3390/ijms26136423_

Round 1

Reviewer 1 Report

Comments and Suggestions for Authors

Line 19: heart failure is not a disease but a general state of many etiologies.

Line 40: there are many forms of cell death than apoptosis or necrosis.

Line 382: many more chemicals should be mentioned

There is a major shortcoming of this review: non cardiologists write about heart failure which does not belong to gastroenterology. This is evident throughout this paper.

A list of cardiomyopathies should be given.

cirrhotic cardiomyopathy is only a very small percentage.

Part 5: Heart failure is very untidily written and many factors are presented in a haphazard way.

In the treatment part field of you should be given of the currently existing employed therapies of cirrhotic cardiomyopathy.

This article contains a lot of useful data. It could profit of a careful revision

A cardiologist should be consulted.

Author Response

Reviewer 1

Line 19: heart failure is not a disease but a general state of many etiologies.

Reply: the reviewer is correct. Our sentence is “in different cardiac conditions including heart failure” Our sentence is in line with the reviewer’s comment.

Line 40: there are many forms of cell death than apoptosis or necrosis.

Reply: as per the reviewer’s comment, we modified the sentence.

Line 382: many more chemicals should be mentioned.

Reply: As per the reviewer’s comment; we added two more chemicals and modified the sentence.

There is a major shortcoming of this review: non cardiologists write about heart failure which does not belong to gastroenterology. This is evident throughout this paper.

Reply: The reviewer is entitled to his/her opinion. However, cirrhotic cardiomyopathy as one of the complications in cirrhosis, has been studied in the Lee lab for almost 4 decades. Moreover, we have been privileged to learn much about heart failure from close collaborations with many different cardiologists over this time.  Cardiac dysfunction in cirrhotic patients/animal models can mimic the cardiac dysfunction in non-cirrhotic patients/animal models. The diagnostic criteria also are adapted from the diagnostic criteria of non-cirrhotic patients. Therefore, many of the learnings of heart failure between non-cirrhotic and cirrhotic patients/animal models are overlapping. Most recently, Dr Lee has co-edited a major textbook on the interaction between heart and liver, with a cardiologist: CARDIO-HEPATOLOGY: CONNECTIONS BETWEEN HEPATIC AND CARDIOVASCULAR DISEASE. Eds: Taniguchi T, Lee SS. Academic Press, 2022. Would that reviewer kindly consider that even though a person was not formally trained in that subspecialty, through a lifelong career spent in another field of study, and learning from expert cardiologists, courses, congresses and literature reading, that person could eventually become somewhat knowledgeable in that area?

A list of cardiomyopathies should be given.

Reply: as per the reviewer’s comment, a list of cardiomyopathies was added (page 2)

cirrhotic cardiomyopathy is only a very small percentage.

Reply: the reviewer is correct. There are not many publications on the role of apoptosis in cirrhotic cardiomyopathy. Our aim in briefly discussing the available publications on apoptosis in non-cirrhotic cardiac conditions was to better explain the possible role of apoptosis in cirrhotic cardiomyopathy, as they may share many common aspects. Just limiting ourselves to the very scant literature on cirrhotic cardiomyopathy is insufficient.

Part 5: Heart failure is very untidily written and many factors are presented in a haphazard way.

Reply: as per the reviewer’s comment, the heart failure section has been modified.

In the treatment part field of you should be given of the currently existing employed therapies of cirrhotic cardiomyopathy.

Reply: as per the reviewer’s suggestion, this part has been enlarged.

This article contains a lot of useful data. It could profit of a careful revision.

Reply: We appreciate the reviewer’s recognition, and have tried to improve the manuscript.

A cardiologist should be consulted.

Reply: Our lab has been collaborating closely with cardiologists for 4 decades. Please see the long paragraph above about non-cardiologists through a lifetime of study and work, learning enough about basic cardiology to write research papers published in the top journals, which have been peer-reviewed and accepted by cardiologists. This current review paper is already written and we respectfully disagree that consulting with a cardiologist at this time would not materially change anything.

Reviewer 2 Report

Comments and Suggestions for Authors

The manuscript could be better organized,  improved and  renewed.

The apoptosis in cardiomyopathy is widely established, so a better understanding of cell cardiomyocytes death programs such as necroptosis, mitochondrial permeability transition pore (mPTP) necrosis, ferroptosis, and pyroptosis should be detailed discussed.

The manuscript it is not clear enough. Since apoptosis in cardiomyopathy is already discussed, the review should focus on cirrhotic cardiomyopathy. So an introduction on cirrhotic cardiomyopathy should be added.

Moreover, the mechanisms of apoptosis involving in this disease should be better described in detail, adding images.

Furthermore, the role of oxidative stress in cirrhotic cardiomyopathy should be better mechanically describe and consequently also antioxidant therapies in cardiomyopathies.

Author Response

Reviewer 2.

The manuscript could be better organized,  improved and  renewed.

Reply: We improved the manuscript according to the reviewers’ comments/suggestions.

The apoptosis in cardiomyopathy is widely established, so a better understanding of cell cardiomyocytes death programs such as necroptosis, mitochondrial permeability transition pore (mPTP) necrosis, ferroptosis, and pyroptosis should be detailed discussed.

Reply: as per the reviewer’s comment, necroptosis, ferroptosis, and pyroptosis were discussed in more detail.

The manuscript it is not clear enough. Since apoptosis in cardiomyopathy is already discussed, the review should focus on cirrhotic cardiomyopathy. So an introduction on cirrhotic cardiomyopathy should be added.

Reply: as per the reviewer’s comment, an introduction on cirrhotic cardiomyopathy was added (page 2).

Moreover, the mechanisms of apoptosis involving in this disease should be better described in detail, adding images.

Reply: Figure 1 shows the extrinsic and intrinsic pathways of apoptosis. The role of apoptosis in cardiac dysfunction is narrated on page 4. “Wencker and colleagues have shown that only a 0.023% rate of cardiomyocyte apoptosis is sufficient to cause a lethal dilated cardiomyopathy in mice”.

Furthermore, the role of oxidative stress in cirrhotic cardiomyopathy should be better mechanically describe and consequently also antioxidant therapies in cardiomyopathies.

Reply: as per the reviewer’s comment, the role of oxidative stress in cirrhotic cardiomyopathy was added in page 7.

Round 2

Reviewer 2 Report

Comments and Suggestions for Authors

The authors have addressed my comments.